# A general framework for formulating structured variable selection

**Guanbo Wang\*** *gwang@hsph.harvard.edu*
*CAUSALab, Departments of Epidemiology, Harvard T.H. Chan School of Public Health, Boston, Massachusetts, USA*

**Mireille E. Schnitzer** *mireille.schnitzer@umontreal.ca*
*Faculté de pharmacie, Université de Montréal, Montréal, Québec, Canada*
*Département de médecine sociale et préventive, Université de Montréal, Québec, Canada*

**Tom Chen** *tom_chen@harvardpilgrim.org*
*Department of Population Medicine, Harvard Medical School and Harvard Pilgrim Health Care Institute, Boston, Massachusetts, USA*

**Rui Wang** *rwang@hsph.harvard.edu*
*Department of Population Medicine, Harvard Medical School and Harvard Pilgrim Health Care Institute, Boston, Massachusetts, USA*
*Department of Biostatistics, Harvard T. H. Chan School of Public Health, Boston, Massachusetts, USA*

**Robert W. Platt** *robert.platt@mcgill.ca*
*Department of Epidemiology, Biostatistics and Occupational Health, McGill University, Montréal, Québec, Canada*

**Reviewed on OpenReview:** *https://openreview.net/forum?id=cvOpIhQQMN*

## Abstract

In variable selection, a selection rule that prescribes the permissible sets of selected variables (called a "selection dictionary") is desirable due to the inherent structural constraints among the candidate variables. Such selection rules can be complex in real-world data analyses, and failing to incorporate such restrictions could not only compromise the interpretability of the model but also lead to decreased prediction accuracy. However, no general framework has been proposed to formalize selection rules and their applications, which poses a significant challenge for practitioners seeking to integrate these rules into their analyses. In this work, we establish a framework for structured variable selection that can incorporate universal structural constraints. We develop a mathematical language for constructing arbitrary selection rules, where the selection dictionary is formally defined. We demonstrate that all selection rules can be expressed as combinations of operations on constructs, facilitating the identification of the corresponding selection dictionary. We use a detailed and complex example to illustrate the developed framework. Once this selection dictionary is derived, practitioners can apply their own user-defined criteria to select the optimal model. Additionally, our framework enhances existing penalized regression methods for variable selection by providing guidance on how to appropriately group variables to achieve the desired selection rule. Furthermore, our innovative framework opens the door to establishing new $\ell_0$-based penalized regression techniques that can be tailored to respect arbitrary selection rules, thereby expanding the possibilities for more robust and tailored model development.

## 1 Introduction

Variable selection has become an important problem in statistics and data science, especially with large-scale and high-dimensional data becoming increasingly available. Variable selection can be used to identify

covariates that are associated with or predictive of the outcome, remove spurious covariates, improve prediction accuracy, and generate hypotheses for causal inquiries (Guyon & Elisseeff, 2003; Reunanen, 2003; Wasserman & Roeder, 2009; Wang et al., 2020; Siddique et al., 2019; Liu et al., 2022; Bouchard et al., 2022; Wang et al., 2023). General techniques to conduct statistical variable selection include best subset selection, penalized regression, and nonparametric approaches like random forest (Heinze et al., 2018; Chowdhury & Turin, 2020).

When selecting variables for the purpose of developing an interpretable model, understanding and formalizing the structure of covariates can lead to more interpretable variable selection. Covariates may have a structure due to

1. Variable type. For example, when including a categorical variable in a regression model, each non-reference category is represented by a binary indicator. It may be desirable to collectively include or exclude these binary indicators as a group.

2. Variable hierarchy. For example, one may define a hierarchical structure for sets of covariates $\mathbb{A}$ and $\mathbb{B}$ such that if $\mathbb{A}$ is selected, then $\mathbb{B}$ must be selected. One application is interaction selection with strong heredity (Haris et al., 2016), that is "the selection of an interaction term requires the inclusion of all main effect terms." A second application is when one covariate is a descriptor of another, such as medication dose (0-10mL) and medication usage (yes/no).

Such restrictions on the resulting model, which we call "selection rules", can be incorporated into the statistical variable selection process so that the resulting model satisfies the rules. Practitioners can define any selection rule based on their *a priori* knowledge of the covariate structure.

Lasso (Tibshirani, 1996) and best subset selection via optimization (Bertsimas et al., 2016) are two commonly-used approaches that do not restrict the composition of the resulting model. However, a variety of existing variable selection techniques have emerged to accommodate diverse selection rules. For instance, the group Lasso (Yuan & Lin, 2006) can select a group of variables collectively, while the sparse group Lasso (Simon et al., 2013) can perform a bi-level selection. Additionally, the Exclusive Lasso (Campbell & Allen, 2017) excels at within-group selection by ensuring at least one variable is chosen from each group. However, these methods have been developed to respect single, specific types of selection rules. Expanding on this, both overlapping group Lasso (Mairal et al., 2010) and latent overlapping group Lasso (Obozinski et al., 2011) can accommodate a more extensive array of (though not all) selection rules by performing the simultaneous selection or exclusion of overlapping groups of variables.

In this work, we develop a general framework for variable selection that can formally express any selection rule in a mathematical language, which enables us to systematically compile the exhaustive list of possible models (i.e. permissible covariate subsets) corresponding to a given selection rule. One practical use of this exhaustive list of models is that we can directly apply statistical criteria to identify the optimal model in terms of the observed data. We also discuss two potential avenues for future development. First, the proposed framework can guide us in how to effectively group variables to follow complex selection rules with existing penalized regression techniques. Second, given the inherent connection between the $\ell_0$ norm and our definition of selection rules, our work also directly leads to new $\ell_0$-based penalized regression methods that can be tailored to accommodate arbitrary selection rules.

This paper is organized as follows. Section 2 is an overview of the key findings with an illustrative example. In Section 3, we provide a formal introduction to the language used in constructing selection rules, prove that we can express any arbitrary selection rule within our framework, and give formulas for deriving the list of all permissible models for any given rule of arbitrary complexity. We then provide a detailed example to illustrate how to set selection rules and derive the selection dictionary in practice, where nine selection rules are considered in Section 4. Last, we discuss the broader implications of this framework and its potential utility in driving future research advancements.

## 2 Overview

Suppose that we have a set of candidate variables $\mathbb{V}$. We define a selection rule on this set as the selection dependencies among all candidate variables. For example, consider a study where we want to investigate which of the following variables should be included in a model for blood pressure: age ($A$), age squared ($A^2$), and race as a categorical variable with 3 levels, represented by dummy variables $B_1$ and $B_2$. We are also interested in the interaction of age with race ($AB_1, AB_2$). So we have $\mathbb{V} = \{A, A^2, B_1, B_2, AB_1, AB_2\}$. In this example, standard statistical practice requires that the resulting model must satisfy a selection rule defined by the following three conditions: 1) if the interaction is selected, then both the main terms for age and race must be selected, 2) if age squared is selected, then age must be selected, 3) the dummy variables representing race must be collectively selected, and 4) the two categorical interaction terms must also be collectively selected. The combination of these four rules is the selection rule that must be respected.

We next define a selection dictionary as the set of all subsets of $\mathbb{V}$ that respect the selection rule. When we say a dictionary respects a selection rule, we mean the dictionary is congruent to the selection rule in the sense that the selection dictionary contains all (rather than some) subsets of $\mathbb{V}$ that respect the selection rule. Theorem 1 in Section 3 states that every selection rule has a unique dictionary. The dictionary for the above example would be $\{\emptyset$ , $\{A\}$ , $\{B_1, B_2\}$ , $\{A, B_1, B_2\}$ , $\{A, A^2\}$ , $\{A, A^2, B_1, B_2\}$ , $\{A, B_1, B_2, AB_1, AB_2\}$ , $\{A, A^2, B_1, B_2, AB_1, AB_2\}\}$. Despite a total of 64 possible subsets of $\mathbb{V}$, there are only 8 possible models that can be selected under this rule.

We are interested in the general problem of finding a selection dictionary given an arbitrary selection rule. We start by defining unit rules as the building blocks of selection rules. For a given set of candidate variables $\mathbb{V}$ with $\mathbb{F} \subseteq \mathbb{V}$, a unit rule is a selection rule of the form "select a number of variables in $\mathbb{F}$." The unit rule depends on the number of variables that are allowed to be selected from $\mathbb{F}$. In our running example, one unit rule is "select zero or two variables from the set $\mathbb{F} = \{B_1, B_2\}$". This is equivalent to saying that $B_1$ and $B_2$ must be selected together, i.e. select neither or both. We define $\mathbb{C}$ as the set of numbers of variables that are allowed to be selected in the unit rule. In the unit rule we gave above, $\mathbb{C} = \{0, 2\}$.

In Theorem 2, we give a formula for the dictionary congruent to a given unit rule. This formula shows that the dictionary is all unique unions of sets 1) of variables in $\mathbb{F}$ where the number of variables is in $\mathbb{C}$ and 2) of variables outside of $\mathbb{F}$. Applying this formula, we can see that the unit rule "select zero or two variables (that is, $\mathbb{C} = \{0, 2\}$) from the set $\mathbb{F} = \{B_1, B_2\}$" has a dictionary that is the set incorporating $\emptyset$ and $\{B_1, B_2\}$ and all unions of $\emptyset$ and $\{B_1, B_2\}$ with any of the other elements in $\mathbb{V}$, respectively.

We then define five useful operations on selection rules in Table 2. For example, $\wedge$ being applied to two selection rules indicates that both of the selection rules must be respected. An arrow $\rightarrow$ indicates if the selection rule on the left-hand side is being respected, then the selection rules on the right-hand side must be respected. For each operation, we can show how the operation on selection rules is related to an operation on the respective dictionaries. Therefore if we are combining or constructing more complex rules from operations on unit rules, we can always derive the resulting dictionary. Our most important result is Theorem 3, stating that any rule can be obtained through operations on unit rules.

To illustrate these ideas in our running example, define unit rules 1) $\mathfrak{u}_1$: "select zero or two variables in $\{AB_1, AB_2\}$," 2) $\mathfrak{u}_2$: "select zero or two variables in $\{B_1, B_2\}$," 3) $\mathfrak{u}_3$: "select two variables in $\{AB_1, AB_2\}$," 4) $\mathfrak{u}_4$: "select three variables in $\{A, B_1, B_2\}$," 5) $\mathfrak{u}_5$: "select one variable in $\{A^2\}$," 6) $\mathfrak{u}_6$: "select one variable in $\{A\}$". The same selection rule that we defined when we introduced the example can be expressed through operations on these unit rules as: $(\mathfrak{u}_1 \wedge \mathfrak{u}_2) \wedge (\mathfrak{u}_3 \rightarrow \mathfrak{u}_4) \wedge (\mathfrak{u}_5 \rightarrow \mathfrak{u}_6)$.

## 3 Selection rules and selection dictionaries

In this section, we introduce the mathematical language for expressing selection rules, which enables us to design algorithms to incorporate selection dependencies into model selection. Two fundamental concepts are being introduced first: the selection rule and its dictionary. Then, we introduce unit rules and operations on unit rules as the building blocks of selection rules. We show that we can construct any selection rule from unit rules and also derive the unit dictionary from set operations on the dictionaries belonging to the unit

rules. We then investigate some properties of the resulting abstract structures. Finally, we discuss some challenges of the direct application of the theory in high-dimensional covariate spaces.

Unless specified otherwise, we use normal math (for example, $F$), blackboard bold ($\mathbb{F}/\mathbb{f}$), Fraktur lowercase ($\mathfrak{f}$), and calligraphy uppercase fonts ($\mathcal{F}$) to represent a random variable, set, rule, and operator respectively. $\mathcal{P}(\mathbb{F})$ represents the power set (collection of all possible subsets) of $\mathbb{F}$, $\mathcal{P}^2(\mathbb{F})$ denotes the power set of the power set of $\mathbb{F}$, and $|\mathbb{F}|$ represents the cardinality of $\mathbb{F}$. The maximum integer of a set of integers $\mathbb{F}$ is denoted by $\max(\mathbb{F})$. We say two sets are equivalent if they contain the same elements, regardless of their multiplicity. For example, $\{A, A, B, B, C, C\} = \{A, B, C\}$. Graphs are helpful to show the dependencies among candidate covariates. For example, an arrow in a graph can indicate that the children nodes are constructed based on their parent nodes.

We take two examples to illustrate the concepts throughout this section.

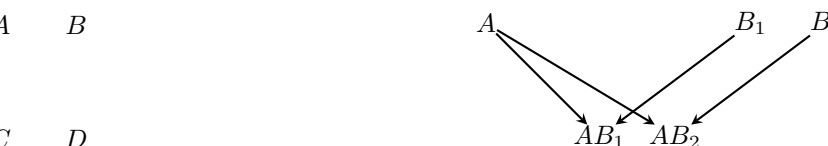

Figure 1: Graph for Example 1          Figure 2: Graph for Example 2

**Example 1.** Suppose that we have 4 candidate variables, $\mathbb{V} = \{A, B, C, D\}$, that have no structural relationship. The corresponding graph is shown in Figure 1.

**Example 2.** Suppose we have three variables: a continuous variable $A$ and a three-level categorical variable $B$ (represented by two dummy indicators $B_1$ and $B_2$). We also consider their interactions represented by $AB_1$ and $AB_2$. The corresponding graph is shown in Figure 2 with nodes $\mathbb{V} = \{A, B_1, B_2, AB_1, AB_2\}$. The arrows indicate that the child nodes are derived from their parents.

Next, we introduce the concept of the selection rule.

**Definition 1** (Selection rule). *A **selection rule** $\mathfrak{r}$ of $\mathbb{V}$ is defined as selection dependencies among the variables in $\mathbb{V}$.*

The selection dependency is a general concept regarding limitations on which combinations of variables are allowed to be selected into a model. Table 1 gives some examples of selection rules for a covariate set $\mathbb{V} = \{A, B, C\}$.

Table 1: Examples of selection rules and their dictionaries, $\mathbb{V} = \{A, B, C\}$

| $\mathfrak{r}_i$ | **Selection dependencies** | $\mathbb{D}_{\mathfrak{r}_i}$ **Selection dictionaries** |
|---|---|---|
| $\mathfrak{r}_1$ | Select at least one variable in $\{A, B\}$. | $\{\{A\}, \{B\}, \{A, B\}, \{A, C\}, \{B, C\}, \{A, B, C\}\}$ |
| $\mathfrak{r}_2$ | If $A$ is selected, then $B$ must be selected. | $\{\emptyset, \{B\}, \{A, B\}, \{C\}, \{B, C\}, \{A, B, C\}\}$ |
| $\mathfrak{r}_3$ | Respect both $\mathfrak{r}_1$ and $\mathfrak{r}_2$. | $\{\{B\}, \{A, B\}, \{B, C\}, \{A, B, C\}\}$ |

There may be many possible subsets of variables that respect a given selection rule. We define the set of all possible subsets of $\mathbb{V}$ respecting a given selection rule as the corresponding selection dictionary.

Before introducing selection dictionary, we define a general dictionary first.

**Definition 2** (Dictionary). *Given a finite set of candidate variables $\mathbb{V}$, a **dictionary** $\mathbb{D} \subseteq \mathcal{P}(\mathbb{V})$ of $\mathbb{V}$ is a set of subset(s) of $\mathbb{V}$.*

For example, a dictionary of candidate variables $\mathbb{V} = \{A, B, C, D\}$ can be $\{\{A\}, \{B\}\}$, $\{\emptyset, \{A, B, C, D\}, \{A\}\}$ or $\mathcal{P}(\mathbb{V})$ etc.

**Definition 3** (Selection dictionary). *For a given $\mathbb{V}$, a **selection dictionary** $\mathbb{D}_{\mathfrak{r}}$ is a dictionary that contains all subsets of $\mathbb{V}$ that respect the selection rule $\mathfrak{r}$.*

The definition stresses that all sets in a selection dictionary must be a subset of $\mathcal{P}(\mathbb{V})$. This allows us to list all "allowable" sets of variables that could result from a variable selection process respecting the selection rule. Some examples of selection dictionaries corresponding to selection rules are shown in Table 1.

It is also possible to have selection rules that are *contradictory/incoherent*, meaning that they require more variables to be selected than the number of variables in the given set. In this case, we define the selection dictionary to be the $\emptyset$. For instance, if the selection rule is "select 3 variables from $\{A, B\}$," with $\mathbb{V} = \{A, B\}$, then the corresponding selection dictionary is $\emptyset$. When a selection rule is, for example, "select 0 variables from $\{A, B\}$," which is coherent but trivial, then the selection dictionary is the empty set $\{\emptyset\}$.

By the definitions above, there is a mapping from a selection rule to a selection dictionary. The theorem below gives the uniqueness of the mapping with proof in Appendix A.

**Theorem 1.** *Given a selection rule on a set, there is a unique selection dictionary that satisfies this given selection rule.*

We say the unique selection dictionary is *congruent* to its selection rule, which is denoted by $\mathbb{D}_{\mathfrak{r}} \cong \mathfrak{r}$, or equivalently, $\mathfrak{r} \cong \mathbb{D}_{\mathfrak{r}}$. In our context, it is equivalent to saying a selection dictionary respects a selection rule. However, it is possible that more than one selection rule results in the same selection dictionary. Therefore, we define an equivalence class of selection rules below.

**Definition 4** (Equivalence class of selection rules). *For a given candidate set $\mathbb{V}$, and given a selection rule $\mathfrak{r}_1$ with selection dictionary $\mathbb{D}$, the **equivalence class** of $\mathfrak{r}_1$, denoted by $\mathbb{R} := \{\mathfrak{r} : \mathfrak{r} \cong \mathbb{D}\}$, is a set of all selection rules in $\mathbb{V}$ that are congruent to the same selection dictionary.*

**Corollary 1.** *By Definition 4 and Theorem 1, there is a one-to-one mapping from an equivalence class of a selection rule to a selection dictionary.*

For a given (finite) $\mathbb{V}$, we define $\mathfrak{R}$ as the **selection rule space** containing all equivalence classes of selection rules on $\mathbb{V}$. Because the number of possible combinations of selected variables is finite, the number of possible dictionaries is finite. Thus, because of the one-to-one correspondence between dictionaries and rules, the space of rules $\mathfrak{R}$ is also finite.

The above definitions provide us with a broad view of the language for expressing selection rules generally. Next, we introduce the grammar of this language, which allows for the exploration of theoretical properties of selection rules and of further algorithmic development. We start by defining unit rules and their dictionaries and then introduce the operations between unit rules. Then, more complex selection rules can be assembled by unit rules and their operations, and the related selection dictionaries can be determined.

**Definition 5** (Unit rule and its dictionary). *Define $\mathbb{C}$, a set of numbers. For a given $\mathbb{V}$ and a given $\mathbb{F} \subseteq \mathbb{V}$, a **unit rule** $\mathfrak{u}_{\mathbb{C}}(\mathbb{F})$ is a selection rule, where the selection dependency takes the form "select $\mathbb{C}$ variables from $\mathbb{F}$", and the number of variables to be selected is any value in $\mathbb{C}$. A **unit dictionary** $\mathbb{D}_{\mathfrak{u}}$ is a dictionary that contains all subsets of $\mathbb{V}$ that respect the unit rule $\mathfrak{u}_{\mathbb{C}}(\mathbb{F})$.*

**Remark 1.** *The set $\mathbb{C}$ is a set of numbers which constrains the number of variables to be selected in $\mathbb{F}$. For example, if $|\mathbb{F}| = 3$ and $\mathbb{C}$ is $\{1\}$, then the rule in $\mathfrak{u}_{\mathbb{C}}(\mathbb{F})$ translates to "there is one variable to be selected" from $\mathbb{F}$. If $\mathbb{C} = \{0, 2\}$, the unit rule is "there are zero or two variables to be selected" from $\mathbb{F}$. Any $\mathbb{C}$ with elements greater than $|\mathbb{F}|$ would result in an incoherent unit rule because the variable selection is conducted without replacement and so we cannot select more than the cardinality of the set.*

In the context where we investigate more than one unit rule for a given $\mathbb{V}$, we use $\mathfrak{u}_i$ to represent $\mathfrak{u}_{\mathbb{C}_i}(\mathbb{F}_i)$ and number them $i = 1, 2, \dots$. Among the examples in Table 1, only $\mathfrak{r}_1$ is a valid unit rule, which can be expressed as $\mathfrak{u}_{\{1,2\}}(\{A, B\})$, and the unit dictionary $\mathbb{D}_{\mathfrak{u}} = \mathbb{D}_{\mathfrak{r}_1}$ is given.

Given a unit rule and its dictionary, there is a one-to-one mapping from the $\mathbb{F}$ in the unit rule to the corresponding unit dictionary, which is characterized by a unit function $f_{\mathbb{C}}$.

**Definition 6** (Unit function). *Each unit rule relates to a **unit function** $f_{\mathbb{C}}$ with some input $\mathbb{F} \subseteq \mathbb{V}$. A unit function $f_{\mathbb{C}}$ maps the subset $\mathbb{F}$ to the unit dictionary $\mathbb{D}_{\mathfrak{u}} \in \mathcal{P}^2(\mathbb{V})$ that respects $\mathfrak{u}_{\mathbb{C}}(\mathbb{F})$.*

Therefore, for a given unit rule, we can write $\mathfrak{u}_{\mathbb{C}}(\mathbb{F}) \cong \mathbb{D}_{\mathfrak{u}} = f_{\mathfrak{c}}(\mathbb{F})$. This is a valid function because a unit dictionary is defined as the set of all possible subsets of $\mathbb{V}$ that respect the unit rule; thus, given a fixed constraint and set $\mathbb{F}$, there is a unique dictionary output.

The following theorem characterizes unit functions, providing a formula for the unit dictionary so that when a unit rule is given on $\mathbb{F}$, the corresponding unit dictionary can be determined.

**Theorem 2.** *Each unit function in $\mathbb{V}$ is a $\mathbb{C}$-specific function $f_{\mathbb{C}}(\cdot)$, with domain $\mathcal{P}(\mathbb{V})$, defined by*

$$\mathbb{F} \mapsto \mathbb{D}_{\mathfrak{u}} = \left\{ \begin{array}{l} \{\mathfrak{a} \cup \mathfrak{b} : \forall \mathfrak{a} \subseteq \mathbb{F} \ s.t. \ |\mathfrak{a}| \in \mathbb{C}, \forall \mathfrak{b} \subseteq \mathbb{V} \setminus \mathbb{F}\}, \ if \ |\mathbb{F}| \geqslant \max(\mathbb{C}) \\ \emptyset, \ otherwise, \end{array} \right.$$

*where $\mathbb{F} \in \mathcal{P}(\mathbb{V})$.*
*When $|\mathbb{F}| \geqslant \max(\mathbb{C})$, the unit rule $\mathfrak{u}_{\mathbb{C}}(\mathbb{F})$ is coherent, and the unit function is a bijection with domain $\mathbb{M} = \{\mathbb{F}, \ s.t. \ \mathbb{F} \in \mathcal{P}(\mathbb{V}), |\mathbb{F}| \geqslant \max(\mathbb{C})\}$ and image $\{\mathfrak{a} \cup \mathfrak{b}, \forall \mathfrak{a} \subseteq \mathbb{F} \ s.t. \ |\mathfrak{a}| \in \mathbb{C}, \forall \mathfrak{b} \subseteq \mathbb{V} \setminus \mathbb{F}, \forall \mathbb{F} \in \mathbb{M}\}$.*

This means that when the unit rule is coherent, the unit dictionary contains all sets that are unions between a subset of $\mathbb{F}$ that respects the constraint $\mathbb{C}$ and a subset of the remaining covariates in $\mathbb{V}$ (excluding $\mathbb{F}$). The proof is given in Appendix B. Corollary 2 gives a special case of Theorem 2, characterizing the mapping of $\mathbb{V}$ to a unit dictionary by a unit function. Corollary 3 gives an interesting property of a unit dictionary. The proofs are direct consequences of Theorem 2.

**Corollary 2.** *When the input of a unit function is $\mathbb{V}$, with a constraint $\mathbb{C}$ resulting in a coherent unit rule, the resulting unit dictionary is $f_{\mathbb{C}}(\mathbb{V}) = \{\mathfrak{n} \in \mathcal{P}(\mathbb{V}) : |\mathfrak{n}| \in \mathbb{C}\}$.*

**Corollary 3.** *When $\mathbb{C} \neq \{0\}$ for a given coherent unit rule, the corresponding unit dictionary $\mathbb{D}_{\mathfrak{u}}$ satisfies $\cup_i \mathbb{D}_{\mathfrak{u},i} = \mathbb{V}$, where $\mathbb{D}_{\mathfrak{u},i}$ is the ith set in $\mathbb{D}_{\mathfrak{u}}$.*

To further investigate the relationships among unit functions with different constraints, we provide the following corollaries.

**Corollary 4.** *For a given $\mathbb{F} \in \mathcal{P}(\mathbb{V})$, $f_{(.)}(\mathbb{F})$ is injective with respect to the argument $\mathbb{C}$ when at least one constraint ($\mathbb{C}_1$ or $\mathbb{C}_2$) results in a coherent rule applied to $\mathbb{F}$. That is, $f_{\mathbb{C}_1}(\mathbb{F}) \neq f_{\mathbb{C}_2}(\mathbb{F})$ whenever $\mathbb{C}_1 \neq \mathbb{C}_2$. This means that two distinct unit functions (related to two distinct unit rules) will result in different dictionaries even when the inputs are the same.*

**Corollary 5.** *For $\mathbb{C} = \{0, \ldots, |\mathbb{F}|\}$ then $f_{\mathbb{C}}(\mathbb{F}) = \mathcal{P}(\mathbb{V}), \forall \mathbb{F} \subseteq \mathbb{V}$. This means that when there is effectively no constraint on the selection (i.e. any number of variables can be selected), the unit dictionary is the power set of $\mathbb{V}$. A consequence is that two different unit functions with nonrestrictive constraints can result in the same dictionary even when the inputs are different.*

The proofs of corollaries 4 and 5 are in Appendix C and D, respectively.

The goal is to build selection rules out of unit rules. This will allow for an algorithm to determine the resulting selection dependencies and dictionary. To do this, we define some operations among selection rules. Because a unit rule is also a selection rule, the operations can be applied to unit rules.

**Definition 7** (Operations on selection rules). *Given selection rules on $\mathbb{V}$, define an **operation on selection rules** $\mathcal{O}$ as a function that maps a single selection rule or pair of selection rules to another selection rule $\mathfrak{r}_{\mathcal{O}}$.*

The rule $\mathfrak{r}_{\mathcal{O}}$ resulting from the operation is congruent to a unique selection dictionary which is congruent to $\mathfrak{r}_{\mathcal{O}}, \mathbb{D}_{\mathcal{O}}$. We define five operations in Table 2.

Given an operation on rules, we can derive the corresponding operation on the related dictionaries that will result in the selection dictionary $\mathbb{D}_{\mathcal{O}}$. Table 2 shows the resulting dictionary for each operation. The derivation of each result is given in Appendix E. These results allow us to develop algorithms to output selection dictionaries for complex rules through operations on simpler rules.

We use the running example in Table 1 to illustrate the second and fourth operations on unit rules as a special case.

Table 2: Operations for selection rules and the resulting selection dictionaries.

| Operation | Interpretation | $\mathbb{D}_{\mathcal{O}}$ |
|---|---|---|
| $\neg \mathfrak{r}_1$ | $\mathfrak{r}_1$ is not being respected | $\mathcal{P}(\mathbb{V}) \setminus \mathbb{D}_{\mathfrak{r}_1}$ |
| $\mathfrak{r}_1 \wedge \mathfrak{r}_2$ | both $\mathfrak{r}_1$ and $\mathfrak{r}_2$ are being respected | $\mathbb{D}_{\mathfrak{r}_1} \cap \mathbb{D}_{\mathfrak{r}_2}$ |
| $\mathfrak{r}_1 \vee \mathfrak{r}_2$ | either $\mathfrak{r}_1$ or $\mathfrak{r}_2$, or both is/are being respected | $\mathbb{D}_{\mathfrak{r}_1} \cup \mathbb{D}_{\mathfrak{r}_2}$ |
| $\mathfrak{r}_1 \to \mathfrak{r}_2$ | if $\mathfrak{r}_1$ is being respected, then $\mathfrak{r}_2$ is being respected | $(\mathcal{P}(\mathbb{V}) \setminus \mathbb{D}_{\mathfrak{r}_1}) \cup (\mathbb{D}_{\mathfrak{r}_1} \cap \mathbb{D}_{\mathfrak{r}_2})$ |

Selection rule $\mathfrak{r}_i$ on $\mathbb{V}$ is congruent to $\mathbb{D}_{\mathfrak{r}_i}, i = 1, 2$.

Define
$$\mathfrak{u}_1 := \mathfrak{u}_{\{1,2\}}(\{A, B\}) \cong \mathbb{D}_{\mathfrak{u}_1} = \{\{A\}, \{B\}, \{A, B\}, \{A, C\}, \{B, C\}, \{A, B, C\}\},$$
$$\mathfrak{u}_2 := \mathfrak{u}_{\{1\}}(\{A\}) \cong \mathbb{D}_{\mathfrak{u}_2} = \{\{A\}, \{A, B\}, \{A, C\}, \{A, B, C\}\},$$
$$\mathfrak{u}_3 := \mathfrak{u}_{\{1\}}(\{B\}) \cong \mathbb{D}_{\mathfrak{u}_3} = \{\{B\}, \{A, B\}, \{B, C\}, \{A, B, C\}\}.$$

The $\mathfrak{r}_2$ in Table 1 is "if $A$ is selected, then $B$ must be selected," which can be expressed as $\mathfrak{r}_2 := \mathfrak{u}_2 \to \mathfrak{u}_3$. According to Table 2, $\mathfrak{r}_2$ is congruent to $\{\emptyset, \{B\}, \{A, B\}, \{C\}, \{B, C\}, \{A, B, C\}\}$, which is exactly the $\mathbb{D}_{\mathfrak{r}_2}$ in Table 1.

Note that, by Definition 1, the operation on two selection rules results in a selection rule, thus the results of an operation on two selection rules can be an input of a second operation. We use parentheses to differentiate the order of operations. The $\mathfrak{r}_3$ in Table 1 is "select at least one variable in $\{A, B\}$" $\wedge \mathfrak{r}_2$. Thus, $\mathfrak{r}_3 := \mathfrak{u}_1 \wedge (\mathfrak{u}_2 \to \mathfrak{u}_3)$ is a valid operation resulting in a rule that is congruent to the selection dictionary $\{\{B\}, \{A, B\}, \{B, C\}, \{A, B, C\}\}$ (according to Table 2), which is exactly the $\mathbb{D}_{\mathfrak{r}_3}$ in Table 1.

We provide some useful properties of operations below, which can be verified by checking the resulting dictionaries for both sides of the equations. These properties can be used to identify which selection rules are in the same equivalence class.

**Proposition 1.** *Given $\mathfrak{r}_1 \neq \mathfrak{r}_2 \neq \mathfrak{r}_3$ (in the sense that the congruent dictionaries are distinct), then*

1. *Commutative laws: $\mathfrak{r}_1 \wedge \mathfrak{r}_2 = \mathfrak{r}_2 \wedge \mathfrak{r}_1$; $\mathfrak{r}_1 \vee \mathfrak{r}_2 = \mathfrak{r}_2 \vee \mathfrak{r}_1$,*

2. *Associative laws: $(\mathfrak{r}_1 \wedge \mathfrak{r}_2) \wedge \mathfrak{r}_3 = \mathfrak{r}_1 \wedge (\mathfrak{r}_2 \wedge \mathfrak{r}_3)$; $(\mathfrak{r}_1 \vee \mathfrak{r}_2) \vee \mathfrak{r}_3 = \mathfrak{r}_1 \vee (\mathfrak{r}_2 \vee \mathfrak{r}_3)$,*

3. *Non-distributive laws: $\mathfrak{r}_1 \vee (\mathfrak{r}_2 \wedge \mathfrak{r}_3) \neq (\mathfrak{r}_1 \vee \mathfrak{r}_2) \wedge (\mathfrak{r}_1 \vee \mathfrak{r}_3)$;*
   $\mathfrak{r}_1 \wedge (\mathfrak{r}_2 \vee \mathfrak{r}_3) \neq (\mathfrak{r}_1 \wedge \mathfrak{r}_2) \vee (\mathfrak{r}_1 \wedge \mathfrak{r}_3)$, *and*

4. *Sequential laws $(\mathfrak{r}_1 \to \mathfrak{r}_2) \wedge (\mathfrak{r}_1 \to \mathfrak{r}_3) = \mathfrak{r}_1 \to (\mathfrak{r}_2 \wedge \mathfrak{r}_3)$;*
   $(\mathfrak{r}_1 \to \mathfrak{r}_2) \vee (\mathfrak{r}_1 \to \mathfrak{r}_3) = \mathfrak{r}_1 \to (\mathfrak{r}_2 \vee \mathfrak{r}_3)$,*

*all apply.*

The next theorem confirms that, equipped with operations and unit rules, we can now effectively express any selection rule as operations on unit rules.

**Theorem 3.** *All selection rules can be expressed by either unit rules or operations on unit rules using $\wedge$ and $\vee$.*

The proof is given in Appendix F. This means that we can represent any rule in a mathematical language. This allows us to develop algorithms to combine multiple rules and generate resulting dictionaries.

Next, we use Examples 1 and 2 with a hypothetical data structure to illustrate how to express some common selection dependencies by unit rules and operations. The corresponding selection dictionaries are also provided.

**Example 1.1** (Individual selection) In Example 1, suppose all variables are continuous or binary, and no structure is imposed. We can set the selection rule as selection between 0 to 4 variables $\mathfrak{r} = \mathfrak{u}_{\{0,1,2,3,4\}}(\mathbb{V})$, and then $\mathbb{D}_{\mathfrak{r}} = \mathcal{P}(\mathbb{V})$. This rule is satisfied by the (adaptive) Lasso (Tibshirani, 1996).

**Example 1.2** (Groupwise selection) In Example 1, suppose we have 2 three-level categorical variables. Denote $\mathbb{F}_1 = \{A, B\}, \mathbb{F}_2 = \{C, D\}$. Let the variables in $\mathbb{F}_1$ be the dummy variables representing a categorical variable, and similarly for the variables in $\mathbb{F}_2$. In an analysis, we would like to select $\mathbb{F}_1$ collectively, same for $\mathbb{F}_2$. We can then set $\mathfrak{r} = \mathfrak{u}_{\{0,2\}}(\mathbb{F}_1) \wedge \mathfrak{u}_{\{0,2\}}(\mathbb{F}_2)$. In addition, $\mathbb{D}_{\mathfrak{r}} = \{\emptyset, \mathbb{F}_1, \mathbb{F}_2, \mathbb{F}_1 \cup \mathbb{F}_2\}$. This rule is satisfied by the group Lasso (Yuan & Lin, 2006).

**Example 1.3** (Within-group selection) If variables in $\mathbb{F}_1 = \{A, B\}$ are one group, and $\mathbb{F}_2 = \{C, D\}$ represents a second group, and the goal is to select at least one variable from both groups, (Campbell & Allen, 2017; Kong et al., 2014) then we set $\mathfrak{r} = \mathfrak{u}_{\{1,2\}}(\mathbb{F}_1) \wedge \mathfrak{u}_{\{1,2\}}(\mathbb{F}_2)$, meaning there is at least one variable that must be selected in $\mathbb{F}_1$ and $\mathbb{F}_2$ respectively. In addition, $\mathbb{D}_{\mathfrak{r}} = \big\{\{A, C\}, \{B, C\}, \{A, B, C\}, \{A, D\}, \{B, D\}, \{A, B, D\}, \{A, C, D\}, \{B, C, D\}, \{A, B, C, D\}\big\}$. This rule is satisfied by the exclusive (group) Lasso (Campbell & Allen, 2017).

**Example 2.1** (Categorical interaction selection with strong heredity) In Example 2, because $\{B_1, B_2\}$ are dummy variables representing the same categorical variable, they have to be collectively selected. Similarly for $\{AB_1, AB_2\}$. In addition, there is a common rule that is being applied in interaction selection, which is called strong heredity (Haris et al., 2016; Lim & Hastie, 2015): "if the interaction is selected, then all of its main terms must be selected". Define $\mathfrak{u}_1 = \mathfrak{u}_{\{0,2\}}\{B_1, B_2\}, \mathfrak{u}_2 = \mathfrak{u}_{\{0,2\}}\{AB_1, AB_2\}, \mathfrak{u}_3 = \mathfrak{u}_{\{2\}}\{AB_1, AB_2\}$, $\mathfrak{u}_4 = \mathfrak{u}_{\{3\}}\{A, B_1, B_2\}$. The selection rule $\mathfrak{r} = (\mathfrak{u}_1 \wedge \mathfrak{u}_2) \wedge (\mathfrak{u}_3 \rightarrow \mathfrak{u}_4)$ satisfies the common selection dependencies imposed for categorical interaction selection and strong heredity. In addition, $\mathbb{D}_{\mathfrak{r}} = \{\emptyset, \{A\}, \{B_1, B_2\}, \{A, B_1, B_2\}, \{A, B_1, B_2, AB_1, AB_2\}\}$. This rule can be satisfied by the overlapping group Lasso (Jenatton et al., 2011; Yuan et al., 2011).

**Example 2.2** (Categorical interaction selection with weak heredity) Another common rule that can be applied to interaction selection is weak heredity (Haris et al., 2016): "if the interaction is selected, then at least one of its main terms must be selected". To write weak heredity in terms of operations on unit rules, we further define $\mathfrak{u}_5 = \mathfrak{u}_{\{1,3\}}\{A, B_1, B_2\}$. Then with the unit rules defined in Example 2.1, the selection rule $\mathfrak{r} = (\mathfrak{u}_1 \wedge \mathfrak{u}_2) \wedge (\mathfrak{u}_3 \rightarrow \mathfrak{u}_5)$ satisfies the common selection dependencies imposed for categorical interaction selection under weak heredity. In addition, the corresponding selection dictionary is the union of the $\mathbb{D}_{\mathfrak{r}}$ in Example 2.1 and $\{\{A, AB_1, AB_2\}, \{B_1, B_2, AB_1, AB_2\}\}$. This rule is satisfied by the latent overlapping group Lasso (Obozinski et al., 2011).

Thus, using Theorem 2 and the set operations indicated in Table 2, one can directly construct a selection dictionary. Alternatively, one can obtain the selection dictionary by exhaustively checking whether each possible subset of variables satisfies the desired selection rule. For instance, suppose $\mathbb{V} = \{A, B, C\}$ and $\mathfrak{r} = \mathfrak{u}_1(\{A\}) \rightarrow \mathfrak{u}_1(\{B\})$, then we can check if each $\varkappa \in \mathcal{P}(\mathbb{V})$ satisfies the condition $\{\{A, B\} \in \varkappa \text{ or } \{A\} \notin \varkappa\}$. If $\varkappa$ satisfies the condition, then it should be included in the selection dictionary. Encoding the conditions in a programming language requires the use of both set and logic operations.

For practical purposes, exhaustive checking can be more prone to errors as it requires manual encoding of each selection rule. On the other hand, direct construction can be automated with generic functions. Moreover, the computational efficiency of these approaches can vary depending on the programming language used. If a language is more suited to set operations than logic operations, exhaustive checking might be less efficient. This is because exhaustive checking relies heavily on logic operations within the programming process.

However, the applicability of the proposed framework is limited when dealing with high-dimensional candidate predictors. The limitation arises because both the direct construction and exhaustive checking approaches necessitate generating and manipulating the power set of candidate predictors. Handling an enormous power set on a local computer may not be feasible. However, in cases where the selection rules apply to only a low-dimensional set of covariates, it is feasible to construct the selection dictionary differently. This can be achieved by merging the sub-selection dictionary, which is derived from the low-dimensional set of covariates, with the power set of the remaining covariates. This method simplifies the process in such specific scenarios and lowers the computational burden.

## 4 An illustrative example

To enhance the understanding and applicability of the proposed framework, we give a specific application where multiple selection rules are to be applied in the construction of an interpretable prediction model.

We consider a setting where we want to anticipate major bleeding among patients hospitalized for atrial fibrillation who are initiating oral anticoagulants (OAC). We want a prediction model that can be understood by clinical users for both credibility and usefulness in clinical decision-making. Qazi et al. (2021) investigated a similar prediction problem without incorporating any selection rules. Using similar example, Wang et al. (2024a) and Wang et al. (2024b) constructed coherent prediction models

OAC includes warfarin, which is administered in a continuous dose depending on biological response, and direct oral anticoagulants (DOAC) encompassing apixaban, dabigatran, and rivaroxaban, each available in both high and low doses. Patients are prescribed one of these OAC, to be taken regularly. One important hypothesis is that patient adherence to their medication prescription predicts treatment outcomes. A patient's adherence profile can be inferred from patterns in the use of other chronic-use medications, like hypertension drugs. For example, if a patient consistently uses their prescribed hypertension medication, it is typically indicative of high adherence to their other prescribed medication(s).

For our purposes, suppose the potential predictors in the analysis include age, sex, $CHA_2DS_2$-VASc score (a stroke risk score for patients with atrial fibrillation), comorbidities, OAC type used at cohort entry, concomitant medications (including antiplatelets, nonsteroidal anti-inflammatory drugs (NSAIDs), antidepressants, and proton pump inhibitors (PPIs)), and the drug-drug interactions between OAC type and concomitant medications. Let `DOAC` indicate that a patient took a DOAC (instead of warfarin). Let `Apixaban` and `Dabigatran` indicate whether a patient took the drug, respectively. Let `High-dose-DOAC` indicate whether a patient took a high-dose of a DOAC, where `High-dose-DOAC`$= 0$ means that the patient was either receiving low-dose-DOAC or warfarin. Let `High-adherence` indicate that the patient adhered to another chronic medication prescription.

The covariates above suggest the usage of selection rules in order to arrive at a meaningful model. For example, in a prediction model, including interaction terms without their main effects compromises its statistical interpretability. Additionally, some covariates have nested relationships – for example, `Apixaban` is nested in `DOAC`, only possibly taking a value of one if `DOAC`$= 1$. Consequently, without including `DOAC` in the model, the coefficient of `Apixaban` is not easily interpretable.

We next outline nine key selection rules, which are described symbolically in Table 3. Selection rule 1 forces `DOAC` into the model if `High-dose-DOAC` is included. This ensures that the coefficient of `High-dose-DOAC` (if selected) can be interpreted as the contrast of high-dose-DOAC versus low-dose-DOAC, which is of interest. This rule is important because if `High-dose-DOAC` is selected without `DOAC`, then its coefficient interpretation would be the contrast of high-dose-DOAC versus low-dose-DOAC and warfarin combined, offering limited clinical insights. This rationale also applies to rules 2 and 3, which also force the inclusion of the higher-level indicator (`DOAC`) when the nested indicator (`Apixaban` or `Dabigatran`) is included. Rules 4 and 6-9 involve strong heredity, i.e. both main terms must be included if an interaction term is included. Rule 5 is formulated based on a combination of these factors.

Thus, the selection rule is the combination of these 9 individual selection rules through the $\wedge$ operation. After establishing a selection rule, the next step involves deriving a corresponding selection dictionary according to Table 2. First, a selection dictionary is identified for each of the nine selection rules. These rules are all "if-then" rules. Take selection rule 2 as an example: the permissible combination for `Apixaban` and `DOAC` is $\{\emptyset, \{\texttt{DOAC}\}, \{\texttt{Apixaban}, \texttt{DOAC}\}\}$. This rule imposes no constraints on other variables, so every selection dictionary element for rule 2 combines an element from the power set of other variables with one from the set $\{\emptyset, \{\texttt{DOAC}\}, \{\texttt{Apixaban}, \texttt{DOAC}\}\}$. To synthesize all selection rules, intersect the selection dictionaries from each of the nine rules.

The resulting selection dictionary is useful in guiding variable grouping in penalized regression methods, such as the latent overlapping group Lasso, which is discussed in the following section.

Table 3: The selection rules for the illustrative example.

| # | Selection Rule |
|---|---|
| 1 | $\mathfrak{u}_{\{1\}}(\{\texttt{High-dose-DOAC}\}) \rightarrow \mathfrak{u}_{\{1\}}(\{\texttt{DOAC}\})$ |
| 2 | $\mathfrak{u}_{\{1\}}(\{\texttt{Apixaban}\}) \rightarrow \mathfrak{u}_{\{1\}}(\{\texttt{DOAC}\})$ |
| 3 | $\mathfrak{u}_{\{1\}}(\{\texttt{Dabigatran}\}) \rightarrow \mathfrak{u}_{\{1\}}(\{\texttt{DOAC}\})$ |
| 4 | $\mathfrak{u}_{\{1\}}(\{\texttt{DOAC} \times \texttt{High-adherence}\}) \rightarrow \mathfrak{u}_{\{2\}}(\{\texttt{DOAC, High-adherence}\})$ |
| 5 | $\mathfrak{u}_{\{1\}}(\{\texttt{High-dose-DOAC} \times \texttt{High-adherence}\}) \rightarrow \mathfrak{u}_{\{4\}}(\{\texttt{DOAC, High-adherence,}$ $\texttt{High-dose-DOAC, DOAC} \times \texttt{High-adherence}\})$ |
| 6 | $\mathfrak{u}_{\{1\}}(\{\texttt{DOAC} \times \texttt{Antiplatelets}\}) \rightarrow \mathfrak{u}_{\{2\}}(\{\texttt{DOAC, Antiplatelets}\})$ |
| 7 | $\mathfrak{u}_{\{1\}}(\{\texttt{DOAC} \times \texttt{NSAIDs}\}) \rightarrow \mathfrak{u}_{\{2\}}(\{\texttt{DOAC, NSAIDs}\})$ |
| 8 | $\mathfrak{u}_{\{1\}}(\{\texttt{DOAC} \times \texttt{Antidepressants}\}) \rightarrow \mathfrak{u}_{\{2\}}(\{\texttt{DOAC, Antidepressants}\})$ |
| 9 | $\mathfrak{u}_{\{1\}}(\{\texttt{DOAC} \times \texttt{PPIs}\}) \rightarrow \mathfrak{u}_{\{2\}}(\{\texttt{DOAC, PPIs}\})$ |

## 5 Discussion

Covariate structures often exhibit intricate complexities in real-world data analysis, and incorporating those complex structures into variable selection is instrumental in enhancing both model interpretability and prediction accuracy. Previous efforts in this domain have typically tackled the issue by developing penalized regression methods that either incorporate a specific selection rule or impose a particular grouping structure, yet none have offered a comprehensive solution to address the problem in its full generality.

This manuscript is a first step in addressing this research gap. Our framework allows us to define generic selection rules through a universal mathematical formulation. Furthermore, we have introduced the formal link between any arbitrary rule and its corresponding selection dictionary, which is the space of all permissible covariate subsets that respect the selection rule. Our derivation of the properties of these mathematical objects allowed us to establish these relationships and identify avenues for future development.

The developed framework offers several valuable applications. Firstly, the resulting selection dictionary can be employed directly in low-dimensional scenarios to select the optimal model among all permissible models. This selection process can be guided by user-defined criteria, such as goodness-of-fit metrics like AIC or BIC, or prediction accuracy measures like cross-validated prediction error. It is important to note that manually enumerating the elements of the selection dictionary is a labor-intensive task prone to errors, making our framework a significant time-saving and error-reducing solution.

Secondly, given a penalized regression method, such as the (latent) overlapping group Lasso, which is a penalized regression method that can handle multiple selection rules, an existing gap in the literature is a general approach to identifying the grouping structure that respects a given selection rule. The developed framework enables us first to express the complex selection rule, and then use the corresponding selection dictionary to guide us in how to group variables. We address this strategy for building overlapping group Lasso grouping structures in greater detail in ongoing work.

Thirdly, current penalized regression methods tailored for structured variable selection contain limitations on the selection rules they can satisfy. In particular, they do not allow for a restriction on the number of covariates from a set allowed into the model, which we defined as our unit rule. For example, the (latent) overlapping group Lasso is unable to respect the selection rule $\mathfrak{u}_{\{0,2\}}(\{A, B, C\})$. This limitation stems from its inability to dictate the number of variables included in the selected model, as this depends on the data. In other words, it cannot guarantee that the selected model will contain a specific number of variables, as required by the rule. However, we can alternatively consider using the $\ell_0$ norm, which counts the non-zero elements in a vector, in a penalized regression. A unit rule $\mathfrak{u}_{\{1,2\}}\{A, B, C, D\}$ necessitates selecting fewer than three variables from the set $\{A, B, C, D\}$. If we define $\boldsymbol{\beta}$ as the vector of coefficients of the variables in this set, this unit rule can be translated into $\|\boldsymbol{\beta}\|_0 \leqslant 2$, which can be introduced as a constraint in a penalized regression. According to Theorem 3, operations on such constraints enable us to derive a constraint for any arbitrary selection rule. In light of these capabilities and considering the recent advancements in algorithms

for solving $\ell_0$-norm penalized regressions (Bertsimas et al., 2016), the next steps of our work will develop an $\ell_0$ norm-based penalized regression based on our framework. This will allow for the incorporation of completely general selection rules into variable selection.

We anticipate that by addressing the research gaps identified in this study, we will see an increase in real-world applications that explicitly define and effectively incorporate selection rules of increasing complexity, leading to more interpretable prediction models. These models will offer deeper insights into prediction mechanisms, proving valuable and beneficial for domain-specific users. The enhanced interpretability is expected to make these models more appreciated and widely utilized in their respective fields.

The proposed framework unifies the structured variable selection problem and creates a paradigm where researchers can view the problem generically rather than starting from a specific class of covariate structure and rule, ignoring all others. Generic guidance for variable selection rules would allow practitioners to scrutinize the covariate structures in their application carefully and potentially incorporate a larger scope of desirable selection rules. As the landscape of data and applications continues to evolve, the emergence of novel selection rules is inevitable. Our framework is purposefully designed with adaptability at its core, ensuring its capability to seamlessly integrate these emerging rules and be a useful resource for future applications.

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

# A    Proof of Theorem 1

*Proof.* Suppose $\mathbb{D}_{\mathfrak{r},1}$ and $\mathbb{D}_{\mathfrak{r},2}$ respect the same selection rule $\mathfrak{r}$. Denote a subset of $\mathbb{V}$ by $\mathfrak{d}$. If $\mathfrak{d} \in \mathbb{D}_{\mathfrak{r},1}$, then $\mathfrak{d} \in \mathbb{D}_{\mathfrak{r},2}$ by Definition 3. Without loss of generality, now suppose $\mathfrak{d} \notin \mathbb{D}_{\mathfrak{r},1}$, then by Definition 3, $\mathfrak{d}$ does not respect $\mathfrak{r}$, so $\mathfrak{d} \notin \mathbb{D}_{\mathfrak{r},2}$. Therefore, $\mathbb{D}_{\mathfrak{r},1} = \mathbb{D}_{\mathfrak{r},2}$. Therefore, there is a unique dictionary for a given selection rule. $\square$

# B    Proof of Theorem 2

*Proof.* When $|\mathbb{F}| < \max(\mathbb{C})$ then $\mathfrak{u}_{\mathbb{C}}(\mathbb{F})$ is incoherent and the resulting unit dictionary is defined as the $\emptyset$.

When $\mathfrak{u}_{\mathbb{C}}(\mathbb{F})$ is a coherent unit rule, suppose $\mathfrak{d} \in \mathcal{P}(\mathbb{V})$ respects $\mathfrak{u}_{\mathbb{C}}(\mathbb{F})$. Let $\mathfrak{a} = \mathfrak{d} \cap \mathbb{F} \subseteq \mathbb{F}$ such that $|\mathfrak{a}| \in \mathbb{C}$. Let $\mathfrak{b} = \mathfrak{d} \cap (\mathbb{V} \setminus \mathbb{F})$. Then $\mathfrak{d} = \mathfrak{a} \cup \mathfrak{b} \in \{\mathfrak{a} \cup \mathfrak{b}, \forall \mathfrak{a} \subseteq \mathbb{F} \ s.t. \ |\mathfrak{a}| \in \mathbb{C}, \forall \mathfrak{b} \subseteq \mathbb{V} \setminus \mathbb{F}\}$.

Now suppose $\mathfrak{d} \in \mathcal{P}(\mathbb{V})$ does not respect $\mathfrak{u}_{\mathbb{C}}(\mathbb{F})$. Then $\mathfrak{d} \cap \mathbb{F}$ does not respect $\mathfrak{u}_{\mathbb{C}}(\mathbb{F})$. Necessarily, it means that $|\mathfrak{d} \cap \mathbb{F}| \notin \mathbb{C}$. Therefore $\mathfrak{d} \cap \mathbb{F} \notin \{\mathfrak{a} \cup \mathfrak{b}, \forall \mathfrak{a} \subseteq \mathbb{F} \ s.t. \ |\mathfrak{a}| \in \mathbb{C}, \forall \mathfrak{b} \subseteq \mathbb{V} \setminus \mathbb{F}\}$, which implies $\mathfrak{d} \notin \{\mathfrak{a} \cup \mathfrak{b}, \forall \mathfrak{a} \subseteq \mathbb{F} \ s.t. \ |\mathfrak{a}| \in \mathbb{C}, \forall \mathfrak{b} \subseteq \mathbb{V} \setminus \mathbb{F}\}$.

Now we prove that when $\mathfrak{u}_{\mathbb{C}}(\mathbb{F})$ is a coherent unit rule, the related unit function is a bijection.

We prove by contradiction. Suppose there exists two non-empty sets $\mathbb{F}_1 \neq \mathbb{F}_2$, necessarily respecting $|\mathbb{F}_1| \geqslant \max(\mathbb{C}), |\mathbb{F}_2| \geqslant \max(\mathbb{C})$, such that $f_{\mathbb{C}}(\mathbb{F}_1) = f_{\mathbb{C}}(\mathbb{F}_2)$. Denote $\mathbb{M} = \{\mathfrak{a} \cup \mathfrak{b} : \forall \mathfrak{a} \subseteq \mathbb{F}_1 \ s.t. \ |\mathfrak{a}| \in \mathbb{C}, \forall \mathfrak{b} \subseteq \mathbb{V} \setminus \mathbb{F}_1\}$, and $\mathbb{N} = \{\mathfrak{a} \cup \mathfrak{b} : \forall \mathfrak{a} \subseteq \mathbb{F}_2 \ s.t. \ |\mathfrak{a}| \in \mathbb{C}, \forall \mathfrak{b} \subseteq \mathbb{V} \setminus \mathbb{F}_2\}$. So that $\forall \mathfrak{m} \in \mathbb{M}$, $\mathfrak{m}$ satisfies $|\mathfrak{m} \cap \mathbb{F}_1| \in \mathbb{C}$, and $\forall \mathfrak{n} \in \mathbb{N}$, $\mathfrak{n}$ satisfies $|\mathfrak{n} \cap \mathbb{F}_2| \in \mathbb{C}$. By the previous result, if $f_{\mathbb{C}}(\mathbb{F}_1) = f_{\mathbb{C}}(\mathbb{F}_2)$, then $\mathbb{M} = \mathbb{N}$. If $\mathbb{F}_1 \neq \mathbb{F}_2$, then there exists some non-empty $\mathfrak{x}$ such that $\mathfrak{x} \subseteq \mathbb{F}_1$ and $\mathfrak{x} \nsubseteq \mathbb{F}_2$. Suppose that $|\mathfrak{x}| \geqslant \min(C)$. Then $\exists \mathfrak{y}$ such that $\mathfrak{y} \subseteq \mathfrak{x}$ and $|\mathfrak{y}| = \min(\mathbb{C})$. Such $\mathfrak{y}$ is necessarily an element of $\mathbb{M}$. Because $\mathbb{M} = \mathbb{N}$, $\mathfrak{y}$ is necessarily an element of $\mathbb{N}$. According to the definition of $\mathbb{N}$, $\mathfrak{y} = \mathfrak{a}_1 \cup \mathfrak{b}_1$ where $\mathfrak{a}_1$ satisfies $\mathfrak{a}_1 \subseteq \mathbb{F}_2$ such that $|\mathfrak{a}_1| \in \mathbb{C}$, and $\mathfrak{b}_1 \subseteq \mathbb{V} \setminus \mathbb{F}_2$. So necessarily, $|\mathfrak{a}_1| = \min(\mathbb{C})$ and $\mathfrak{b}_1 = \emptyset$. This contradicts $\mathfrak{y} \nsubseteq \mathbb{F}_2$, because $\mathfrak{y} = \mathfrak{a}_1 \subseteq \mathbb{F}_2$.

Now suppose that $|\mathfrak{x}| < \min(\mathbb{C})$. Because the rule is coherent, there exists $\mathfrak{m}$ such that $\mathfrak{x} \subset \mathfrak{m} \subseteq \mathbb{F}_1$ and $|\mathfrak{m}| = \min(\mathbb{C})$. So $\mathfrak{m} \in \mathbb{M} = \mathbb{N}$. Because $\mathfrak{m} \in \mathbb{N}$, we have $\mathfrak{m} = \mathfrak{a}_2 \cup \mathfrak{b}_2$, and necessarily $|\mathfrak{a}_2| = \min(\mathbb{C})$, so $\mathfrak{b}_2 = \emptyset$ and $\mathfrak{m} = \mathfrak{a}_2 \subseteq \mathbb{F}_2$. Therefore, $\mathfrak{x} \subseteq \mathbb{F}_2$, which contradicts $\mathfrak{x} \nsubseteq \mathbb{F}_2$. $\square$

# C    Proof of corollary 4

*Proof.* Without loss of generality, suppose $\mathfrak{u}_{\mathbb{C}_1}(\mathbb{F})$ is a coherent unit rule, and $\exists c_1 \in \mathbb{C}_1$ such that $c_1 \notin \mathbb{C}_2$. By Theorem 2, $\exists \mathfrak{d} \in f_{\mathbb{C}_1}(\mathbb{F})$ such that $|\mathfrak{m} \cap \mathbb{F}| = c_1$. Then by Theorem 2, because $c_1 \notin \mathbb{C}_2$, $\mathfrak{d} \notin f_{\mathbb{C}_2}(\mathbb{F})$.

$\square$

# D    Proof of corollary 5

*Proof.* By Corollary 2, the property holds when $\mathbb{F} = \mathbb{V}$. Now suppose $\mathbb{F} \subset \mathbb{V}$. By Theorem 2, $f_{\mathbb{C}}(\mathbb{F}) = \{\mathfrak{a} \cup \mathfrak{b}, \forall \mathfrak{a} \subseteq \mathbb{F}, \forall \mathfrak{b} \subseteq \mathbb{V} \setminus \mathbb{F}\}$ when $|\mathbb{F}| \geqslant \max(\mathbb{C})$, which is $\mathcal{P}(\mathbb{V})$. Thus, $f_{\mathbb{C}}(\mathbb{F}) = \mathcal{P}(\mathbb{V}), \forall \mathbb{F} \subseteq \mathbb{V}$. $\square$

# E    Proof of mapping rules on dictionaries

*Proof.* For each operation on rules $\mathfrak{r}_1$ and $\mathfrak{r}_2$ with respective dictionaries $\mathbb{D}_{\mathfrak{r}_1}$ and $\mathbb{D}_{\mathfrak{r}_2}$ in Table 2, we prove that the rule $\mathcal{O}_{\mathfrak{r}}(\mathfrak{r}_1, \mathfrak{r}_2)$ is congruent to the operation on dictionaries in the third column.

1. $\mathcal{O}_{\mathfrak{r}}(\mathfrak{r}_1) = \neg \mathfrak{r}_1$: suppose there is a set $\mathfrak{d} \in \mathcal{P}(\mathbb{V})$ such that it does not respect $\mathfrak{r}_1$. Then by Definition 5, $\mathfrak{d} \in \mathcal{P}(\mathbb{V}) \setminus \mathbb{D}_{\mathfrak{r}_1}$. Now suppose $\mathfrak{d}$ is a set that does respect $\mathfrak{r}_1$. Then $\mathfrak{d} \in \mathbb{D}_{\mathfrak{r}_1}$, and thus $\mathfrak{d} \notin \mathcal{P}(\mathbb{V}) \setminus \mathbb{D}_{\mathfrak{r}_1}$. So, the dictionary congruent to $\neg \mathfrak{r}_1$ is $\mathcal{P}(\mathbb{V}) \setminus \mathbb{D}_{\mathfrak{r}_1}$.

2. $\mathcal{O}_{\mathfrak{r}}(\mathfrak{r}_1, \mathfrak{r}_2) = \mathfrak{r}_1 \wedge \mathfrak{r}_2$: suppose there is a set $\mathbb{d} \in \mathcal{P}(\mathbb{V})$ such that it respects $\mathfrak{r}_1$ and $\mathfrak{r}_2$. Then by Definition 5, $\mathbb{d} \in \mathbb{D}_{\mathfrak{r}_1} \cap \mathbb{D}_{\mathfrak{r}_2}$. Without loss of generality, now suppose $\mathbb{d}$ is a set that does not respect $\mathfrak{r}_1$, then $\mathbb{d} \in \mathcal{P}(\mathbb{V}) \setminus \mathbb{D}_{\mathfrak{r}_1}$, and thus $\mathbb{d} \notin \mathbb{D}_{\mathfrak{r}_1} \cap \mathbb{D}_{\mathfrak{r}_2}$. Thus, the dictionary congruent to $\mathfrak{r}_1 \wedge \mathfrak{r}_2$ is $\mathbb{D}_{\mathfrak{r}_1} \cap \mathbb{D}_{\mathfrak{r}_2}$.

3. $\mathcal{O}_{\mathfrak{r}}(\mathfrak{r}_1, \mathfrak{r}_2) = \mathfrak{r}_1 \vee \mathfrak{r}_2$: suppose there is a set $\mathbb{d} \in \mathcal{P}(\mathbb{V})$ such that it respects $\mathfrak{r}_1$ and/or $\mathfrak{r}_2$. Then by Definition 5, $\mathbb{d} \in \mathbb{D}_{\mathfrak{r}_1} \cup \mathbb{D}_{\mathfrak{r}_2}$. Now suppose $\mathbb{d}$ is a set that respects neither $\mathfrak{r}_1$ nor $\mathfrak{r}_2$, then $\mathbb{d} \in \left(\mathcal{P}(\mathbb{V}) \setminus \mathbb{D}_{\mathfrak{r}_1}\right) \cap \left(\mathcal{P}(\mathbb{V}) \setminus \mathbb{D}_{\mathfrak{r}_2}\right)$, and thus $\mathbb{d} \notin \mathbb{D}_{\mathfrak{r}_1} \cup \mathbb{D}_{\mathfrak{r}_2}$. Thus, the dictionary congruent to $\mathfrak{r}_1 \vee \mathfrak{r}_2$ is $\mathbb{D}_{\mathfrak{r}_1} \cup \mathbb{D}_{\mathfrak{r}_2}$.

4. $\mathcal{O}_{\mathfrak{r}}(\mathfrak{r}_1, \mathfrak{r}_2) = \mathfrak{r}_1 \rightarrow \mathfrak{r}_2$: an arbitrary set $\mathbb{d} \in \mathcal{P}(\mathbb{V})$ falls into one of four categories, 1) $\mathbb{d}$ respects both $\mathfrak{r}_1$ and $\mathfrak{r}_2$, 2) $\mathbb{d}$ respects neither $\mathfrak{r}_1$ nor $\mathfrak{r}_2$, 3) $\mathbb{d}$ respects only $\mathfrak{r}_2$ but not $\mathfrak{r}_1$, and 4) $\mathbb{d}$ respects only $\mathfrak{r}_1$ but not $\mathfrak{r}_2$. A set $\mathbb{d}$ in the first three categories respects $\mathfrak{r}_1 \rightarrow \mathfrak{r}_2$. We first show that sets $\mathbb{d}$ in the first three categories belong to $(\mathcal{P}(\mathbb{V}) \setminus \mathbb{D}_{\mathfrak{r}_1}) \cup (\mathbb{D}_{\mathfrak{r}_1} \cap \mathbb{D}_{\mathfrak{r}_2})$, and a set $\mathbb{d}$ in category 4) does not.

   (a) If $\mathbb{d}$ is in category 1), then $\mathbb{d} \in \mathbb{D}_{\mathfrak{r}_1} \cap \mathbb{D}_{\mathfrak{r}_2}$, which belongs to $(\mathcal{P}(\mathbb{V}) \setminus \mathbb{D}_{\mathfrak{r}_1}) \cup (\mathbb{D}_{\mathfrak{r}_1} \cap \mathbb{D}_{\mathfrak{r}_2})$.
   (b) If $\mathbb{d}$ is in category 2), then $\mathbb{d} \in (\mathcal{P}(\mathbb{V}) \setminus \mathbb{D}_{\mathfrak{r}_1}) \cap (\mathcal{P}(\mathbb{V}) \setminus \mathbb{D}_{\mathfrak{r}_2})$, which belongs to $(\mathcal{P}(\mathbb{V}) \setminus \mathbb{D}_{\mathfrak{r}_1}) \cup (\mathbb{D}_{\mathfrak{r}_1} \cap \mathbb{D}_{\mathfrak{r}_2})$.
   (c) If $\mathbb{d}$ is in category 3), then $\mathbb{d} \in (\mathcal{P}(\mathbb{V}) \setminus \mathbb{D}_{\mathfrak{r}_1}) \cap \mathbb{D}_{\mathfrak{r}_2}$, which belongs to $(\mathcal{P}(\mathbb{V}) \setminus \mathbb{D}_{\mathfrak{r}_1}) \cup (\mathbb{D}_{\mathfrak{r}_1} \cap \mathbb{D}_{\mathfrak{r}_2})$.
   (d) If $\mathbb{d}$ is in category 4), then $\mathbb{d} \in \mathbb{D}_{\mathfrak{r}_1} \cap (\mathcal{P}(\mathbb{V}) \setminus \mathbb{D}_{\mathfrak{r}_2})$, which does not belong to $(\mathcal{P}(\mathbb{V}) \setminus \mathbb{D}_{\mathfrak{r}_1}) \cup (\mathbb{D}_{\mathfrak{r}_1} \cap \mathbb{D}_{\mathfrak{r}_2})$.

   This completes the proof.

   $\square$

# F    Proof of Theorem 3

*Proof.* The theorem is equivalent to saying that for a given rule $\mathfrak{r}$ on $\mathbb{V}$, the related dictionary $\mathbb{D}$ can be obtained by unions and/or intersections of unit dictionaries.

Suppose that the selection dictionary has cardinality 0. Then it is equal to a unit dictionary of an incoherent unit rule.

Now suppose that the selection dictionary is a set with cardinality 1. Let $\mathbb{D}_{\mathfrak{r}} = \{\mathbb{F}\}$, for some $\mathbb{F} \subseteq \mathbb{V}$. Let $\mathbb{D}_{\mathfrak{u}_1}$ and $\mathbb{D}_{\mathfrak{u}_2}$ be dictionaries corresponding to unit rules $\mathfrak{u}_1 = \mathfrak{u}_{\{|\mathbb{F}|\}}(\mathbb{F})$ and $\mathfrak{u}_2 = \mathfrak{u}_{\{0\}}(\mathbb{V} \setminus \mathbb{F})$, respectively. Then $\mathbb{D}_{\mathfrak{r}}$ can be expressed as $\mathbb{D}_{\mathfrak{u}_1} \cap \mathbb{D}_{\mathfrak{u}_2}$. Thus, $\mathfrak{r} = \mathfrak{u}_1 \wedge \mathfrak{u}_2$.

We have demonstrated that we can construct a selection dictionary with a single element using unit dictionaries. Selection dictionaries containing more than one element can be constructed by taking the unions of selection dictionaries with single elements. $\square$

