# OpenReview forum: "A general framework for formulating structured variable selection"
_TMLR — Accepted by TMLR_

### Review · Reviewer_JXAA · 2023-11-13

**Summary Of Contributions:**

This paper contributes by introducing a general framework for structured variable selection, establishing a mathematical language to formalize diverse selection rules. This enhances existing methods and introduces new approaches for variable selection. These contributions provide a more systematic and flexible approach to real-world data analysis, potentially influencing the fields of statistics and data science.

**Audience:**

Yes

**Broader Impact Concerns:**

Upon comprehensive review, no further ethical concerns or broader impact issues have been identified beyond the scope already addressed in the existing statement.

**Claims And Evidence:**

Yes

**Requested Changes:**

1. Which methods can handle multiple variable selection rules? What advantages and disadvantages do these methods present?

2. In the first paragraph of the introduction, citations should precede punctuations.

3. Please check the weaknesses.

**Strengths And Weaknesses:**

Strengths:
1. The paper introduces a general framework, innovating within the domain of structured variable selection, offering a fresh perspective for addressing structural constraints in real-world data.

2. Developing a mathematical language to formalize various selection rules provides a more systematic approach to handling structural constraints.

Weaknesses:
1. The examples presented in the paper might be overly abstract. Including more specific and detailed cases could provide clearer illustrations of the application of the framework and methods.

2. Improving the accuracy of language and terminological descriptions is necessary to precisely express and define terms. Occasionally, sentence structures may obscure information.

3. The paper touches on potential directions for future research, but it could be more specific in indicating the practical impact and contributions these directions might yield.

---

> ### Author Response · Authors · 2023-11-22
> **Reply to Reviewer JXAA**
>
> We are grateful for the reviewer's insightful suggestions and valuable comments. We have integrated all the recommended revisions into the manuscript, which we believe have significantly enhanced the quality of our paper.
>
> `Requested Changes:
> Which methods can handle multiple variable selection rules? What advantages and disadvantages do these methods present?`
>
> We thank the reviewer's question. The (latent) overlapping group Lasso can handle multiple selection rules. In fact, to our best knowledge, it is the method that can handle the most flexible selection rules in penalized regression methods. The advantage is its built-in flexibility regarding the selection rules it can respect (see the change below), and the drawbacks are 1) the lack of a general approach to identifying the grouping structure that respects a given selection rule (as discussed in the fourth paragraph of the discussion), and 2) there are still some selection rules that it cannot respect (see the change below).
>
> We added the following italicized sentences or segments to the discussion.
>
> Secondly, given a penalized regression method, such as the (latent) overlapping group Lasso,  *which is a penalized regression method that can handle multiple selection rules*, an existing gap...
>
> Thirdly, current penalized regression methods tailored for structured variable selection contain limitations on the selection rules they can satisfy. In particular, they do not allow for the restriction on the number of covariates to be selected, which we defined as our unit rule. *For example, the (latent) overlapping group Lasso is unable to respect the selection rule $\mathfrak{u}_{\\{0,2\\}}(\\{A, B, C\\})$. This limitation stems from its inability to dictate the number of variables included in the selected model, as this depends on the data. In other words, it cannot guarantee that the selected model will contain a specific number of variables, as required by the rule.*
>
> `Requested Changes:
> In the first paragraph of the introduction, citations should precede punctuations.`
>
> We thank the reviewer for pointing it out. We have corrected them.
>
> > `Weaknesses:
> The examples presented in the paper might be overly abstract. Including more specific and detailed cases could provide clearer illustrations of the application of the framework and methods.`
>
> We thank the reviewer for the suggestion. In the revised version, we added Section 4 (we did not copy-paste it here due to the space limit), which includes an illustrative and practical example. In the example, we want to construct an interpretable prediction model for the incidence of major bleeding among hospitalized atrial fibrillation patients who initiate oral anticoagulants (OAC), aiming to understand the mechanism of major bleeding and support future clinical decision-making. To do so, nine selection rules are established with justified rationale, followed by the derivation of the selection dictionary
>
> > `Weaknesses:
> Improving the accuracy of language and terminological descriptions is necessary to precisely express and define terms. Occasionally, sentence structures may obscure information.`
>
> We thank the reviewer for closely looking into the writing style. We have scrutinized the paper again and improved the writing where we think it can be improved.
>
> > `Weaknesses:
> The paper touches on potential directions for future research, but it could be more specific in indicating the practical impact and contributions these directions might yield.`
>
> We thank the reviewer's suggestion. We added the following paragraph at the end of the discussion section.
>
> *We anticipate that by addressing the research gaps identified in this study, we will see an increase in real-world applications that explicitly define and effectively incorporate selection rules of increasing complexity, leading to more interpretable prediction models. These models will offer deeper insights into prediction mechanisms, proving valuable and beneficial for domain-specific users. The enhanced interpretability is expected to make these models more appreciated and widely utilized in their respective fields.*
>
> ***In the revised manuscript, we have highlighted all significant changes in blue for easy identification and review.***

---

> ### Comment · Reviewer_JXAA · 2023-11-29
> **Reply to Responses**
>
> Thank you for your thorough and diligent response to the review comments. Your response addressed some of the questions I had regarding the paper.
>
> JXAA

---

### Review · Reviewer_ATiA · 2023-11-15

**Summary Of Contributions:**

The paper outlines the mathematical formalism to encode feature selection rules and a systematic way of enumerating all feature subsets that are satisfying the constraints - selection dictionary. The atomic component of framework is a 'unit rule' and it was shown that by using logical operators to combine unit rules, any selection rule can be described. Moreover, it was shown how can selection dictionary be derived from selection dictionaries corresponding to individual unit rules, using union and intersection operators.

**Audience:**

Yes

**Claims And Evidence:**

Yes

**Requested Changes:**

* From my perspective, it would be very valuable to at least comment on the computational aspect of the framework - as it would shed the light on practical applicability of the proposed framework. For example, due to its "combinatorial" nature, it might be not applicable to very high-dimensional problems.
* It would be good to directly state, potentially with an example, what specific kinds of selection rules the overlapping group LASSO (and latent overlapping group LASSO) cannot cover.
* Maybe it would be good to comment on the selection rules which are contradictory, as no selection dictionary can satisfy such a rule.
* The fifth operation 'exclusive OR' is missing in the Table 2.

**Strengths And Weaknesses:**

Strengths:
* The framework is novel and is generalizing previous specialized instances of structured feature selection
* Framework is precisely described and theorems and corollaries appears correct by directly deriving the properties or proven correct using induction or contradiction
* Text is very well written and conveys the messages smoothly


Weaknesses:
* Unlike the most of the structured feature selection that it is generalizing upon (group LASSO, etc.), this framework is quite computationally expensive due to need to explicitly evaluate all the subset of features from the selection dictionary
* Another concern from computational perspective is need to derive selection dictionaries for individual unit rules before deriving final selection dictionary

---

> ### Author Response · Authors · 2023-11-22
> **Reply to Reviewer ATiA (1/2)**
>
> We thank the reviewer for closely examining our work; we appreciate the efforts and comments, which we believe improve our work a lot. Please see the response below.
>
> `Requested Changes:
> From my perspective, it would be very valuable to at least comment on the computational aspect of the framework - as it would shed the light on practical applicability of the proposed framework. For example, due to its ``combinatorial'' nature, it might be not applicable to very high-dimensional problems.`
>
> We thank the reviewer for providing this comment. In the revised version, at the end of section 3, we added the following paragraphs in the revised version.
>
> *Thus, using Theorem 2 and the set operations indicated in Table 2, one can directly construct a selection dictionary. Alternatively, one can obtain the selection dictionary by exhaustively checking whether each possible subset of variables satisfies the desired selection rule. For instance, suppose $\mathbb{V}=\\{A, B, C\\}$ and $\mathfrak{r}=\mathfrak{u}\_{1}(\\{A\\})\rightarrow\mathfrak{u}_{1}(\\{B\\})$, then we can check if each $\mathbb{x}\in\mathcal{P}(\mathbb{V})$ satisfies the condition $\\{\\{A, B\\}\in\mathbb{x} \texttt{ or } \\{A\\}\notin\mathbb{x}\\}$. If $ \mathbb{x} $ satisfies the condition, then it should be included in the selection dictionary. Encoding the conditions in a programming language requires the use of both set and logic operations.*
>
> *For practical purposes, exhaustive checking can be more prone to errors as it requires manual encoding of each selection rule. On the other hand, direct construction can be automated with generic functions. Moreover, the computational efficiency of these approaches can vary depending on the programming language used. If a language is more suited to set operations than logic operations, exhaustive checking might be less efficient. This is because exhaustive checking relies heavily on logic operations within the programming process.*
>
> *However, the applicability of the proposed framework is limited when dealing with high-dimensional candidate predictors. The limitation arises because both the direct construction and exhaustive checking approaches necessitate generating and manipulating the power set of candidate predictors. Handling an enormous power set on a local computer may not be feasible. However, in cases where the selection rules apply to only a low-dimensional set of covariates, it is feasible to construct the selection dictionary differently. This can be achieved by merging the sub-selection dictionary, which is derived from the low-dimensional set of covariates, with the power set of the remaining covariates. This method simplifies the process in such specific scenarios and lowers the computational burden.*
>
> `Requested Changes:
> It would be good to directly state, potentially with an example, what specific kinds of selection rules the overlapping group LASSO (and latent overlapping group LASSO) cannot cover.`
>
> We appreciate the reviewer's suggestion and have incorporated an example (see below) in our discussion section. This example highlights future research directions, inspired by the limitations of the (latent) overlapping group Lasso in respecting certain rules.
>
> *For example, the (latent) overlapping group Lasso is unable to respect the selection rule $\mathfrak{u}\_{\\{0,2\\}}(\\{A, B, C\\})$. This limitation stems from its inability to dictate the number of variables included in the selected model, as this depends on the data. In other words, it cannot guarantee that the selected model will contain a specific number of variables, as required by the rule.*
>
> `Requested Changes: Maybe it would be good to comment on the selection rules which are contradictory, as no selection dictionary can satisfy such a rule.`
>
> We thank the reviewer for the suggestion. After introducing the concept of the selection dictionary, we added the following sentences.
>
> *It is also possible to have selection rules that are contradictory/incoherent, meaning that they require more variables to be selected than the number of variables in the given set. In this case, we define the selection dictionary to be the $\emptyset$. For instance, if the selection rule is "select 3 variables from $\\{A, B\\}$," with $\mathbb{V}=\\{A,B\\}$, then the corresponding selection dictionary is $\emptyset$.*
>
> `Requested Changes:
> The fifth operation 'exclusive OR' is missing in the Table 2.`
>
> We are grateful to the reviewer for highlighting this issue. We previously mentioned a fifth operation in the second-to-last paragraph preceding Proposition 1. Upon reflection, we recognized that this operation is both uncommon in real-world applications and its inclusion potentially detracts from the paper's readability. Our intention was to omit this operation in our initial submission, but we inadvertently retained the associated illustration. To rectify this, we have removed that specific paragraph in the revised version of our paper.

---

> > ### Author Response · Authors · 2023-11-22
> > **Reply to Reviewer ATiA (2/2)**
> >
> > > `Weakness: Unlike the most of the structured feature selection that it is generalizing upon (group LASSO, etc.), this framework is quite computationally expensive due to need to explicitly evaluate all the subset of features from the selection dictionary.`
> >
> > We thank the reviewer for noticing the computational cost of the proposed method for deriving the selection dictionary. We did acknowledge the limitation and have discussed it at the end of Section 3. Please see the reply to the requested change 1 and the added paragraphs.
> >
> > > `Weakness: Another concern from computational perspective is need to derive selection dictionaries for individual unit rules before deriving final selection dictionary.`
> >
> > We thank the reviewer for closely looking into the way of constructing the selection dictionary. We provided an alternative way (exhaustive checking) to obtain selection dictionaries and compare these two approaches. Please see the reply to the requested change 1 and the added paragraphs.
> >
> > ***In the revised manuscript, we have highlighted all significant changes in blue for easy identification and review.***

---

> > > ### Comment · Reviewer_ATiA · 2023-11-30
> > >
> > > Thanks to the authors for detailed responses and thorough changes in the manuscript, I feel my concerns are addressed satisfactorily.

---

### Review · Reviewer_S2hx · 2023-11-17

**Summary Of Contributions:**

The paper deals with the problem of variable selection in statistics.

**Audience:**

No

**Claims And Evidence:**

Yes

**Requested Changes:**

It should be useful to present a real world application of the intuition in order to prove its validity.

**Strengths And Weaknesses:**

The paper is well written and the concepts have been proved by some theorems included in the paper.

---

> ### Author Response · Authors · 2023-11-22
> **Reply to Reviewer S2hx**
>
> We thank the reviewer for the comments and the compliments!
>
>
> `Requested Changes: It should be useful to present a real world application of the intuition in order to prove its validity.`
>
> We appreciate the reviewer's feedback, which allowed us to expand our explanation.
>
> In the revised version, we added Section 4 (we did not copy-paste it here due to the space limit), which includes an illustrative and practical example. In the example, we want to construct an interpretable prediction model for the incidence of major bleeding among hospitalized atrial fibrillation patients who initiate oral anticoagulants (OAC), aiming to understand the mechanism of major bleeding and support future clinical decision-making. To do so, nine selection rules are established with justified rationale, followed by the derivation of the selection dictionary.
>
> This section does not encompass a real-data analysis but rather a proof-of-concept demonstration, more fitting for our purposes. Using this detailed example, we demonstrated the validity and practicality of our approach. A comprehensive real-data analysis, such as building a prediction model, would necessitate further research. This includes determining a grouping structure that aligns with the established selection rules and dictionary, applicable to techniques like the latent overlapping group Lasso. For more insights, please refer to the discussion section.
>
> ***In the revised manuscript, we have highlighted all significant changes in blue for easy identification and review.***

---

### Decision · Action_Editor_p4xT · 2024-01-15

**Recommendation:** Accept as is

**Comment:**

The authors have satisfactorily addressed the reviewers comments. Reviewers acknowledged this, and suggested to accept the paper in the new updated version.  Apart for typographical issues, such as the missing space in "LetApixaban", the paper can be accepted as is.

**Audience:**

The topic covered by the paper is general enough to be of interest for many individuals in TMLR's audience.

**Claims And Evidence:**

The revised version of the paper supports the main claims of the paper. Limitations of the proposed approach are also properly discussed. Overall, the contribution of the paper is significant for the related area.